# Non-invasive monitoring of neoadjuvant radiation therapy response in soft tissue sarcomas by multiparametric MRI and quantification of circulating tumor DNA—A study protocol

**Alexander Runkel**[1,2]\*, **David Braig**[1,3], **Balazs Bogner**[4], **Adrian Schmid**[1], **Ute Lausch**[1], **Anika Boneberg**[1], **Zacharias Brugger**[5], **Anja Eisenhardt**[1], **Jurij Kiefer**[1], **Thomas Pauli**[6], **Melanie Boerries**[6], **Hannah Fuellgraf**[7], **Konrad Kurowski**[7], **Peter Bronsert**[7,8,9], **Jutta Scholber**[10], **Anca-Ligia Grosu**[10], **Philipp Rovedo**[11], **Fabian Bamberg**[4], **Steffen Ulrich Eisenhardt**[1]◉, **Matthias Jung**[2,4]◉

1 Faculty of Medicine, Department of Plastic and Hand Surgery, Medical Center—University of Freiburg, Freiburg, Germany, 2 Faculty of Medicine, Berta-Ottenstein-Programme, University of Freiburg, Freiburg, Germany, 3 Division of Hand, Plastic and Aesthetic Surgery, University Hospital, Ludwig Maximilian University of Munich, Munich, Germany, 4 Faculty of Medicine, Department of Radiology, Medical Center—University of Freiburg, Freiburg, Germany, 5 Faculty of Medicine, Department of Medicine I, Medical Center—University of Freiburg, Freiburg, Germany, 6 Faculty of Medicine, Institute of Medical Bioinformatics, Medical Center—University of Freiburg, Freiburg, Germany, 7 Faculty of Medicine, Institute of Surgical Pathology, Medical Center—University of Freiburg, Freiburg, Germany, 8 Tumorbank Comprehensive Cancer Center Freiburg, Medical Center—University of Freiburg, Freiburg, Germany, 9 Core Facility for Histopathology and Digital Pathology, Medical Center—University of Freiburg, Freiburg, Germany, 10 Faculty of Medicine, Department of Radiation Oncology, Medical Center—University of Freiburg, Freiburg, Germany, 11 Faculty of Medicine, Department of Radiology, Medical Physics, Medical Center-University of Freiburg, Freiburg, Germany

◉ These authors contributed equally to this work.
\* alexander.runkel@uniklinik-freiburg.de

**Data Availability Statement:** All relevant data are within the paper and its Supporting Information files.

## Abstract

### Background

Wide resection remains the cornerstone of localized soft-tissue sarcomas (STS) treatment. Neoadjuvant radiation therapy (NRT) may decrease the risk of local recurrences; however, its effectiveness for different histological STS subtypes has not been systematically investigated. The proposed prospective study evaluates the NRT response in STS using liquid biopsies and the correlation of multiparametric magnetic resonance imaging (mpMRI) with histopathology and immunohistochemistry.

### Methods

Patients with localized high-grade STS, who qualify for NRT, are included in this study.

### Liquid biopsies

Quantification of circulating tumor DNA (ctDNA) in patient blood samples is performed by targeted next-generation sequencing. Soft-tissue sarcoma subtype-specific panel

**Funding:** Author AR received funding from Forschungskommission Freiburg (Grant number: 3095120035). Authors AR and MJ are part of the Berta-Ottenstein-Programme for Clinician Scientists, Faculty of Medicine, University of Freiburg. The funders had and will not have a role in study design, data collection and analysis, decision to publish, or preparation of the manuscript.

**Competing interests:** The authors have declared that no competing interests exist.

sequencing in combination with patient-specific exome sequencing allows the detection of individual structural variants and point mutations. Circulating free DNA is isolated from peri-therapeutically collected patient plasma samples and ctDNA quantified therein. Identification of breakpoints is carried out using FACTERA. Bioinformatic analysis is performed using samtools, picard, fgbio, and the MIRACUM Pipeline.

## mpMRI

Combination of conventional MRI sequences with diffusion-weighted imaging, intravoxel-incoherent motion, and dynamic contrast enhancement. Multiparametric MRI is performed before, during, and after NRT. We aim to correlate mpMRI data with the resected specimen's macroscopical, histological, and immunohistochemical findings.

## Results

Preliminary data support the notion that quantification of ctDNA in combination with tumor mass characterization through co-registration of mpMRI and histopathology can predict NRT response of STS.

## Clinical relevance

The methods presented in this prospective study are necessary to assess therapy response in heterogeneous tumors and lay the foundation of future patient- and tumor-specific therapy concepts. These methods can be applied to various tumor entities. Thus, the participation and support of a wider group of oncologic surgeons are needed to validate these findings on a larger patient cohort.

## Introduction

Sarcomas are rare, malignant tumors of mesenchymal origin and account for approximately 1% of all malignancies (incidence 4-5/100.000) [1]. Sarcomas can occur at virtually every anatomical site, are very heterogeneous and, thus, are clinically challenging. The World Health Organization (WHO) characterizes more than fifty sarcoma subtypes that can occur in almost all age groups [2]. The most frequent soft tissue sarcomas (STS) subtypes are liposarcomas (fat tissue), leiomyosarcomas (smooth muscle), pleomorphic sarcomas, and synovial sarcomas (both highly dedifferentiated tissue of unknown origin), all of which most commonly appear at the extremities and trunk. STS often present as painless swellings and thus remain unrecognized for a long time. Magnetic Resonance Imaging (MRI) and histologically examined core needle biopsies (CNB) serve as diagnostic confirmations for STS [3]. At the time of diagnosis, local tumor growth (>5cm) or metastatic spread has often already occurred, leading to a significant reduction in patient quality of life and overall survival [4].

Patients require extensive radiological imaging for surveillance of recurrences after tumor resection. Tumor metastases most commonly occur in the lungs, might already be evident at the time of initial STS diagnosis, and can also occur years after primary tumor resection. Multimodal treatment concepts combine radiation therapy, surgery, chemotherapy, immunotherapy, and targeted therapeutics [5]. Despite this interdisciplinary approach, around 30% of patients with high-grade tumors eventually develop metastatic disease after multimodal

treatment and succumb to their disease [6]. Adjuvant chemotherapy is still subject to controversial debate [7–9]. The accepted standard treatment strategy for localized tumors includes (neo)adjuvant radiation therapy and complete resection with negative margins [9–13]. Since patients with neoadjuvant radiation therapy (NRT) are up to two times more likely to develop wound healing disorders postoperatively [14], it is important to evaluate which STS patient subpopulation and tumor subtype NRT has positive predictive effects. A Canadian population-based study showed that retroperitoneal STS patients treated with neoadjuvant radiation therapy revealed a superior local recurrence-free survival compared to patients with resection alone [15]. However, there is little evidence of the therapy response of STS during the course of NRT and long-term mortality rates [16]. This might be because no reliable methods to monitor tumor activity have been established to date. Histopathological assessment of the neoadjuvant treated and resected tumor specimen remains the only objective criterion to judge therapy response by estimating histological changes (tumor necrosis and fibrosis) [17, 18]. Yet, no baseline necrosis rate before neoadjuvant therapy is procurable and its prognostic value is disputed [19–21]. A British group showed in a feasibility study that multiparametric MRI (mpMRI) can be used to monitor morphological changes in the tumor under NRT, thereby assessing therapy response for different STS [22]. This study serves as a stepping stone to peritherapeutically monitor neoadjuvant therapy response using non-invasive MRI. Large prospective cohort studies are needed to pave the way for evidence-based recommendations for the use of NRT.

By performing a liquid biopsy, the detection of circulating genetic material through next-generation sequencing represents a potent approach for highly sensitive tumor genotyping [23]. A liquid biopsy only utilizes a small amount of blood withdrawn from the peripheral circulation to quantify cell-free DNA (cfDNA) [24]. These represent small pieces of DNA, which are continuously released into the bloodstream by necrotic and/or apoptotic cells [25]. A small amount of cfDNA is circulating tumor DNA (ctDNA) from tumor cells that harbor tumor-specific genomic alterations. Targeted sequencing of ctDNA improves STS detection independent of its anatomic localization and can be performed anytime during the course of treatment. Quantification of ctDNA in the peripheral circulation might even surpass common imaging modalities in its sensitivity and specificity to detect tumors or metastases [26–28]. In previous works, the Department of Plastic and Hand Surgery, Medical Center—University of Freiburg, Faculty of Medicine, Freiburg, Germany, has shown that liquid biopsy can be used to monitor tumor dynamics, predict minimal residual disease and detect genetic tumor heterogeneities [3, 28–34]. The question remains, whether ctDNA can also be used as a biomarker for tumor activity. If it can be shown that the level of ctDNA in a patient's blood sample correlates with tumor activity, then minimal invasive liquid biopsy can be used peritherapeutically to monitor therapy response. We present a prospective study protocol that correlates ctDNA levels with mpMRI, histopathological and immunohistochemical tumor mass characterization, thereby assessing tumor activity.

## Hypothesis

Since patients with NRT are up to two times more likely to develop wound healing disorders postoperatively [16, 35], it is important to evaluate which STS subtype benefits from NRT. Radiosensitivity varies in STS [36]; therefore, it remains questionable if the NRT recommendation based solely on tumor grading represents an adequate therapy concept. We hypothesize, that due to STS heterogeneity, subtypes respond differently to NRT, as shown in absolute ctDNA copy numbers and relative changes during therapy. In some cases, early tumor resection might lower surgical complications and improve survival rates. Pretreatment copy

numbers of ctDNA can be used as a predictive and prognostic biomarker for Ewing sarcomas [37]. We hypothesize that pretreatment ctDNA levels might correlate with overall and event-free survival for STS patients.

## Strategy

Little evidence exists on the therapy response of STS during the course of neoadjuvant therapy. No reliable methods to monitor tumor activity and response to therapy have been established to date. Histopathological assessment of the resected tumor remains the only objective criterion to judge therapy response by estimating the cell necrosis rate [17, 18]. Yet, no baseline necrosis rate before neoadjuvant therapy is procurable and the prognostic value of core needle biopsies is disputed [3, 19–21, 38]. Magnetic resonance imaging can be used to monitor morphological changes in the tumor under NRT [22].

We plan to utilize non-invasive mpMRI before, during, and after NRT and combine imaging data with an immunohistopathological assessment of the resected tumor tissue. With this data, we can characterize the heterogeneous tumor masses and evaluate macroscopical, and histological responses during the course of NRT. In addition, we plan to monitor ctDNA levels during the course of NRT to obtain information on tumor activity on a cellular level. This data will be correlated with the parameters of mpMRI data and immunohistopathological findings. This enables a much-needed systematic evaluation of therapy response on a cellular, macroscopic, and immunohistopathological level during the course of NRT. In the case of non-responding tumors, NRT can be aborted to avoid potential spreading as well as intra- and postoperative complications. This study will serve as a stepping stone for future tumor- and patient-specific therapy concepts to minimize recurrence and surgical complications.

## Study protocol

This study follows the prospective-specimen-collection, retrospective-blinded-evaluation (PRoBE) design [39]. Samples are collected and archived before knowledge of the clinical course and therapy response and evaluated with mpMRI, histopathology, and immunohistochemistry. Diagnostics and treatment will be performed according to Guideline-compliant and standardized protocols of the Comprehensive Cancer Center Freiburg (CCCF). Imaging will be performed systematically using 3-Tesla MRI (MAGNETOM Vida, Siemens Healthineers) at pre-defined intervals before, during, and after NRT. Neoadjuvant radiation therapy lasts five weeks (5 x 2 Gy radiation per week totaling 50 Gy).

One liquid biopsy consists of 9 ml of blood drawn in an EDTA tube. Over the course of NRT and surgery, approximately ten liquid biopsies are obtained (<100ml blood over 3 months). Each patient receives a whole-exome sequencing (WES) of the tumor and matched non-tumorous DNA to create individual patient / tumor-specific enrichment panels allowing a highly sensitive ctDNA detection in the blood samples.

## Study population and sample size

STS patients treated at the CCCF are included in the study after consent. Only newly diagnosed STS patients with a tumor grading of G2 or G3, who qualify for neoadjuvant radiation therapy, are included in the prospective study. Exclusion criteria are low grade STS, dementia, anemia, age < 16, pregnancy, or the inability to consent.

During the past ten years, over 1500 STS patients haven been treated at CCCF of which >300 patients have been operated on in our department. Over 700 tumor and plasma samples have been collected and stored for research purposes at the department's Biobank. Around 50 newly diagnosed STS patients are seen in our department each year, of which >10 patients

meet the inclusion criteria for this prospective study. Twenty-five patients need to be included in this study to achieve a power of 0.7 and a type-I error of 5% according to a one-sample-mean power calculation. We plan to include >30 patients within the next 36 months.

## Timeline

Blood samples are collected before neoadjuvant radiation therapy and during the treatment before each therapy cycle. In addition, blood samples will be collected the day before and three days after surgery as well as at follow-up examinations (2, 4, 8, 24, and 48 weeks) and archived as samples in the FREEZE-Biobank. Multiparametric MRI will be carried out before, during (after the 3rd cycle), and after neoadjuvant radiation therapy. Further MRI or CT staging will be performed at pre-defined biannual follow-up intervals according to the recommendations of the S3-Guidelines for sarcoma therapy (refer to Figs 1 and 2). Patient recruitment began in 2022 and will last around 36 months.

## Multiparametric magnetic resonance imaging (mpMRI)

mpMRI is used for non-invasive imaging evaluation of the tumor. A multiparametric approach is used to distinguish heterogeneous tumor components and quantitatively assess changes as the sarcoma responds to radiation therapy. MR imaging was performed on a 3.0 T MR scanner (Vida, Siemens Healthineers) using a dedicated 18-channel body coil (Siemens Healthineers). Diffusion weighted imaging (DWI) reflects tumor cellularity, intravoxel incoherent motion (IVIM), and dynamic contrast enhancement (DCE) sequences provide information about tumor microcirculation and vascularity and increase the sensitivity in the evaluation of neoadjuvant radiation therapy response [40–42] (Fig 3). Constant parameters in mpMRI acquisition allow for a standardized and reliable quantitative tumor analysis: apparent diffusion coefficient (ADC), IVIM based pseudo diffusion coefficient $D^*$, perfusion fraction $f$, blood flow-related parameter $fD^*$, and DCE derived $K^{trans}$, $v_e$, $k_{ep}$, $v_p$. Furthermore, volumetric measurements of vital and necrotic tumor portions may be of prognostic value [17–21]. mpMRI data are co-registered with histological findings to validate the identified imaging parameters [22].

## Ex-vivo preparation of the resected tissue

After surgery, the surface of the resected tissue specimen is macroscopically examined and documented at the Institute of Surgical Pathology, Medical Center—University of Freiburg, Faculty of Medicine, Freiburg, Germany. All resection margins are ink-dyed for subsequent histological R-classification (Fig 4). Next, multiple native tumor tissue CNBs are taken MRI-guided (BIM-14/20; Innovative Tomography Products GmbH (ITP)). The regions of interest (ROIs) for the CNBs are determined in the *in-vivo* MRI images before surgery. An *ex-vivo* scan of the resected tissue allows the correct positioning of the biopsy needle. A custom-made MRI-compatible apparatus allows the positioning of the needle on the x-, y-, and z-axis (Fig 5). Multiple pre-defined ROIs are targeted to ensure that biopsies are taken from different vital tumor tissue to increase the sensitivity of our lockdown panels and analyze genetic tumor heterogeneity (Figs 6 and 7).

After native tissue CNBs, the tissue specimen is formalin-fixed for up to 72h depending on its size. After fixation, tumor slices are prepared (Figs 8 and 9). Using a co-registration process, the tumor slices are matched with the corresponding in-vivo mpMRI image. The histopathological slices now mirror the *in-vivo* MRI images which allow the correlation of mpMRI data with immunohistopathological findings.

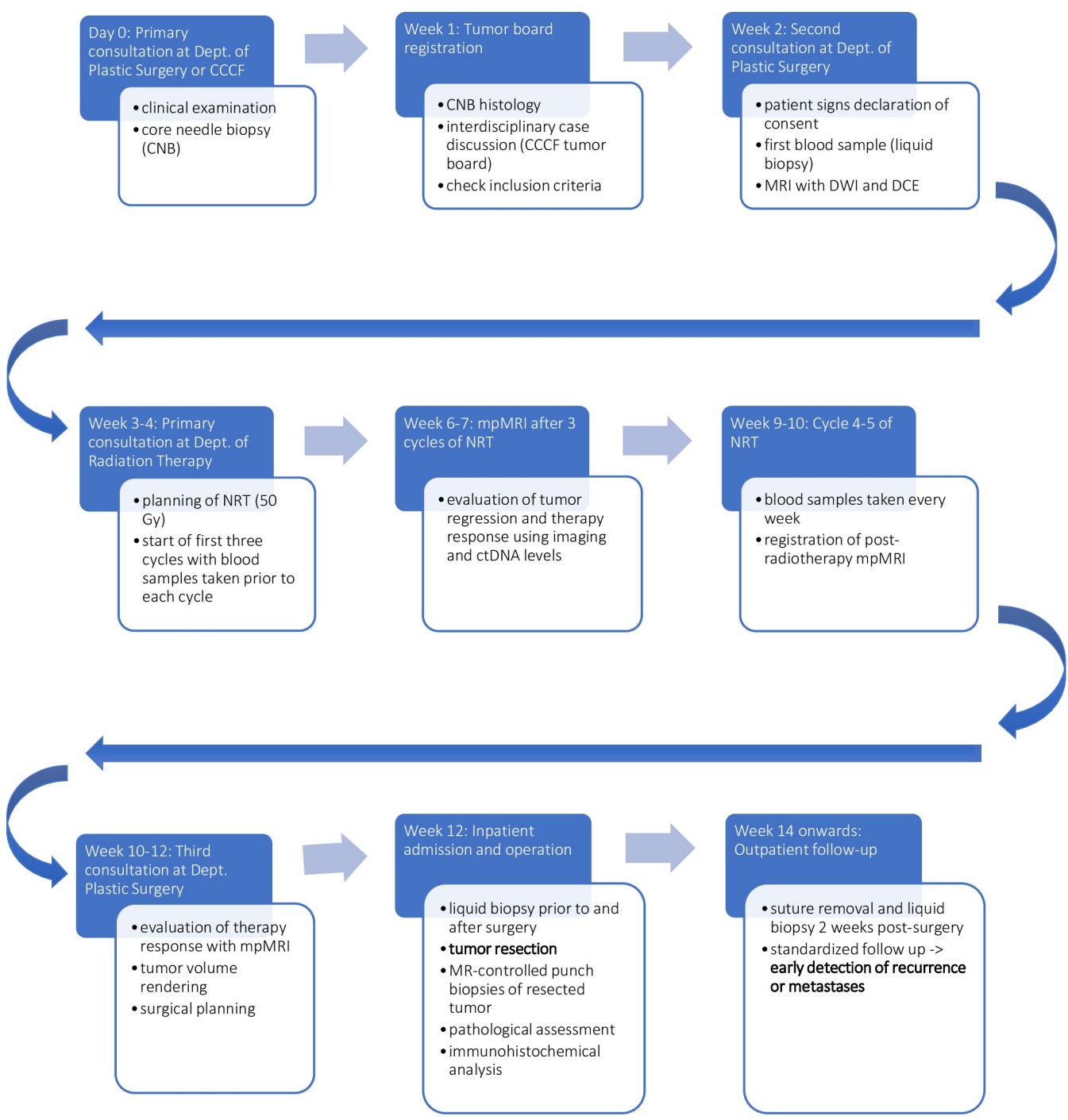

**Fig 1. Timeline of patient diagnostics and therapy steps.**

## Tumor volume rendering and histopathology

Tumor volume and ROI based MRI analyses are performed using the in-house *nora* medical imaging platform (https://www.nora-imaging.com) [43]. A multiparametric tumor analysis is performed at the predefined time intervals before, during, and after therapy (Fig 2). In addition, all histological slides will be stained with hematoxylin and eosin, ALDH1A1 as radiation

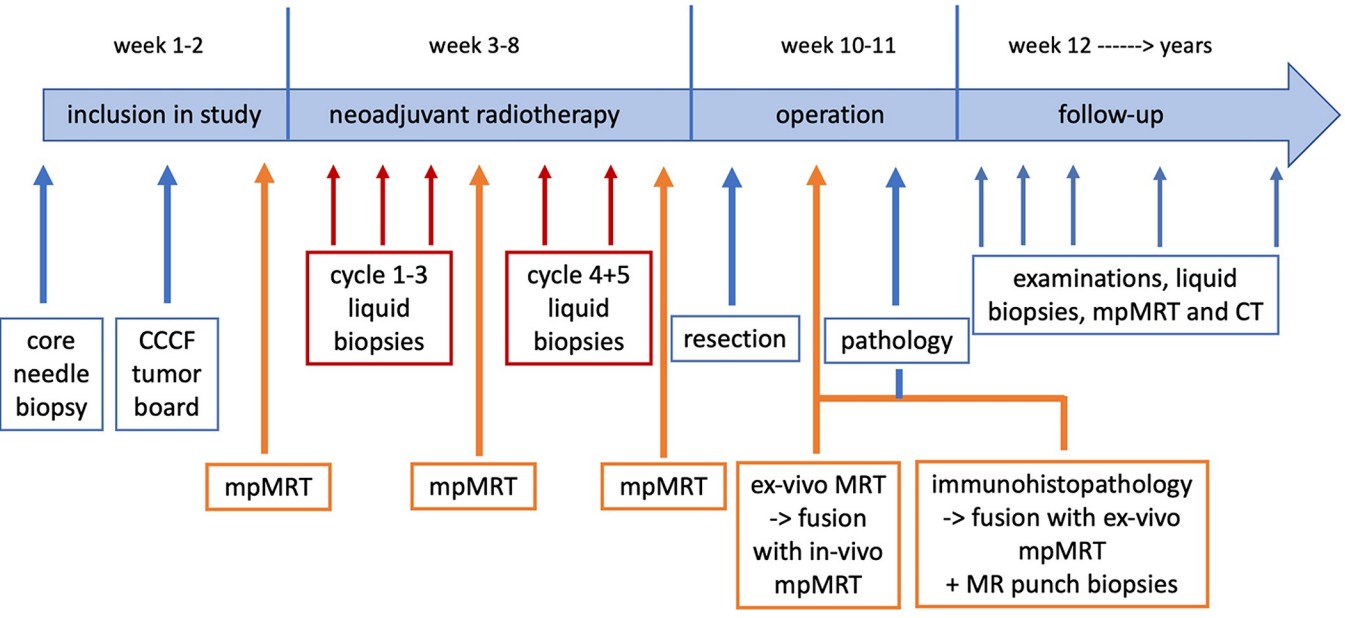

**Fig 2. Timeline of the prospective study protocol.**

resistance, and EMT-/MIB1 as a proliferation marker. All slides are digitalized and co-registered with MRI for correlation analyses. For further workup, all tissue specimens are stored at the CCCF Tumorbank.

## Statistical analysis and sample size calculation

During study planning, we made a sample size calculation in collaboration with the Institute of Medical Biometry and Statistics, IMBI, Freiburg, Germany. We plan to correlate mpMRI parameters (DWI (i.e. ADC), IVIM (i.e. f, $D^*$, and $fD^*$), and DCE (i.e. $K^{trans}$, $v_e$, $k_{ep}$, $v_p$)) with ctDNA levels and immunohistology (cell density proliferation index and ALDH1A1 radiation resistance and MIB1 proliferation marker). We will analyze at least five ROIs (fixed effects) in

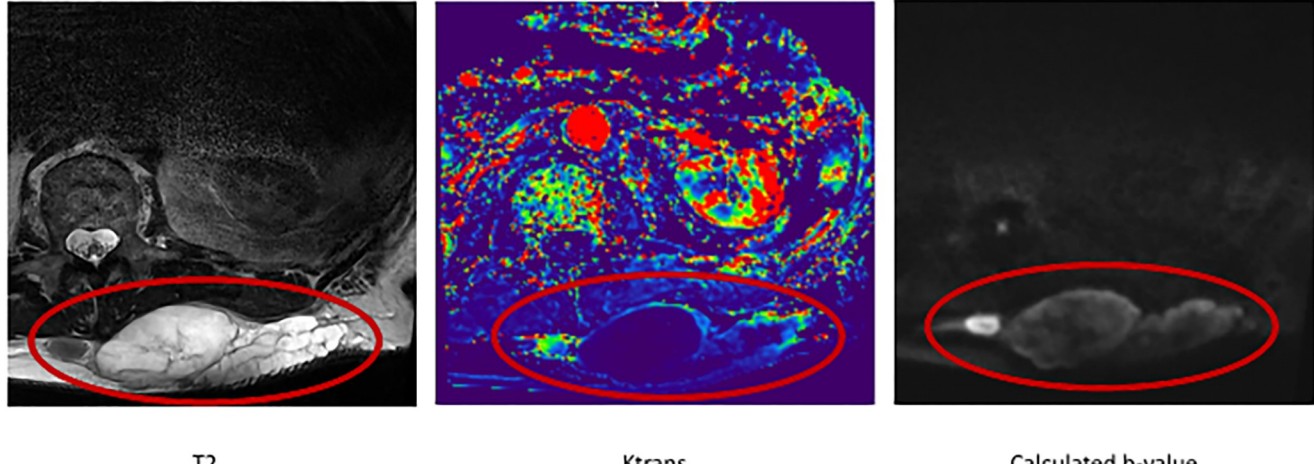

**Fig 3.** T2-weighted MRI (left) with overlayed $K^{trans}$ map (middle) and calculated b-values (on the right) of the in-vivo tumor to identify vital tissue for ex-vivo CNBs.

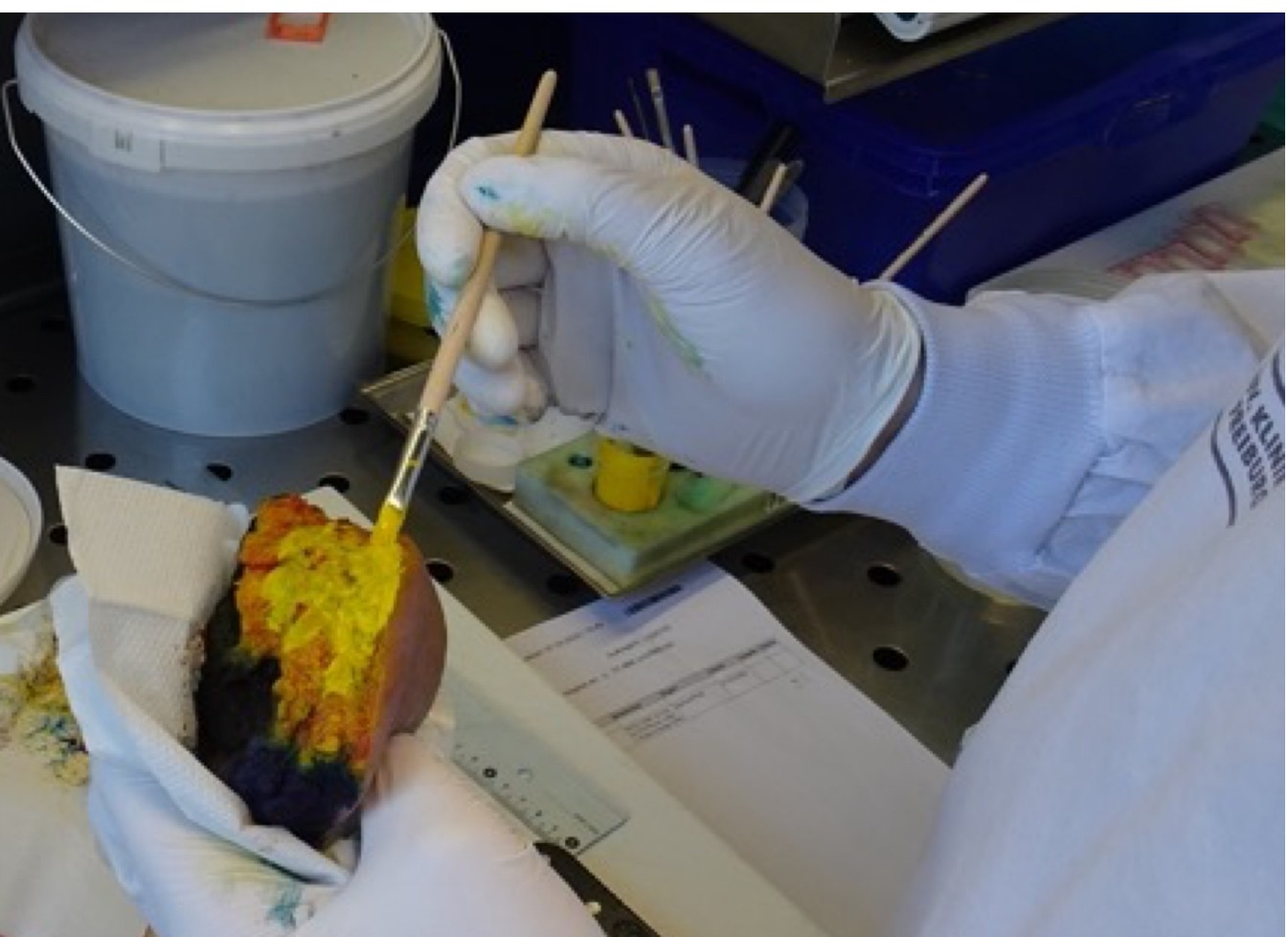

**Fig 4. Staining of the resected tumor is performed by a trained pathologist for evaluation of resection margins.**

each STS patient (random effect), which makes our data nested. Thus, a mixed effects model for nested data is necessary for analyzing associations of quantitative mpMRI parameters with ctDNA levels and (immuno)histology. For modeling with up to five predictors–in our case the quantitative mpMRI sequences–a sample size of 25 patients is necessary for a reliable model.

## Statistical clustering and mpMRI–Histology–ctDNA correlation

The quantitative map values of the DWI (i.e. ADC), IVIM (i.e. f, D*, and fD*), and DCE (i.e. $K^{trans}$, $v_e$, $k_{ep}$, $v_p$) sequence are extracted for each voxel within the tumor ROI. Parametric map values within the tumor ROI are equally weighted for subsequent classification into four groups, based on histological analysis. We perform K-means clustering (Python machine learning library scikit-learn [44]) to split the voxels of the tumor ROI into a set of four groups. The classification and voxel position identified by clustering are mapped onto the T2 sequence by color coding. Clustering is performed using only parametric mpMRI-values. Spatial information is not used as an additional input, but only to identify the clusters on the T2 image. Four representative ROIs—one per cluster–are defined in the co-registered histology and mpMRI section. One-way ANOVA with post hoc pairwise t-tests (Bonferroni-Holm adjusted) is performed to compare differences in mpMRI parameters [45].

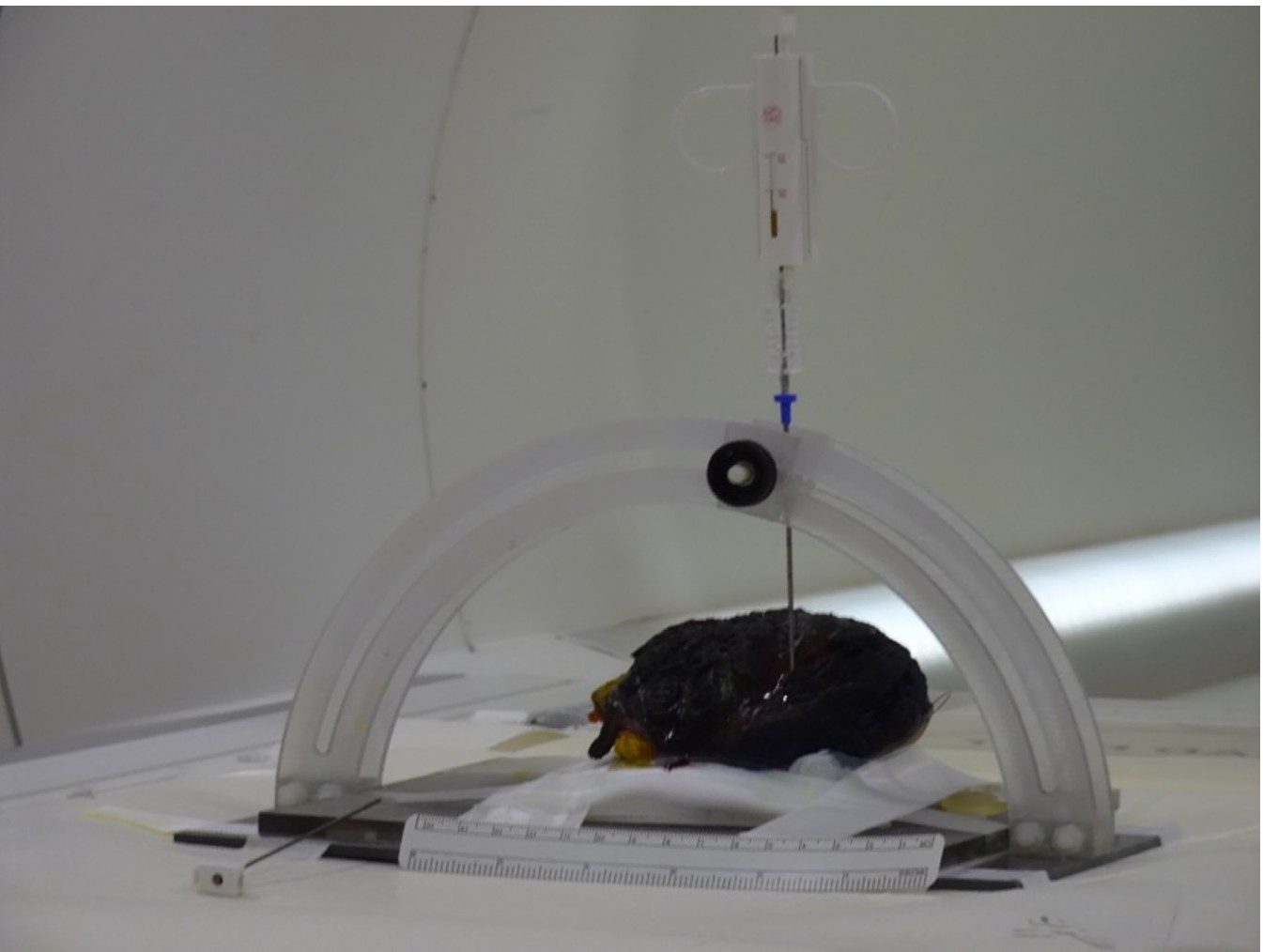

**Fig 5. Custom-made MRI-compatible CNB apparatus allows the precise positioning of the biopsy needle.**

## Methods and bioinformatics

### Blood and tissue sampling

Nine milliliters of whole blood are collected in $K_2EDTA$ tubes, processed within 2 h after withdrawal, and stored in cryotubes (FluidX) at -80°C before use. Plasma samples are stored at the interdisciplinary FREEZE biobank of the University Medical Center Freiburg. Corresponding tumor tissue (formalin-fixed paraffin-embedded (FFPE) and fresh frozen samples) is stored at the CCCF Tumorbank.

### DNA isolation

DNA from tumor native tissue is extracted using DNeasy Blood and Tissue Kit; DNA from FFPE using the Qiagen FFPE Tissue Kit; DNA from blood/leukocytes using the DNeasy Blood and Tissue Kit and cell-free DNA is extracted using the QIAamp Circulating Nucleic Acid Kit. All preparation steps are according to the manufacturer's instructions (Qiagen, Hilden, Germany). The quantity of DNA and cfDNA in plasma is determined by Qubit 3.0 Fluorometer, and dsDNA HS reagents (Invitrogen, Carlsbad).

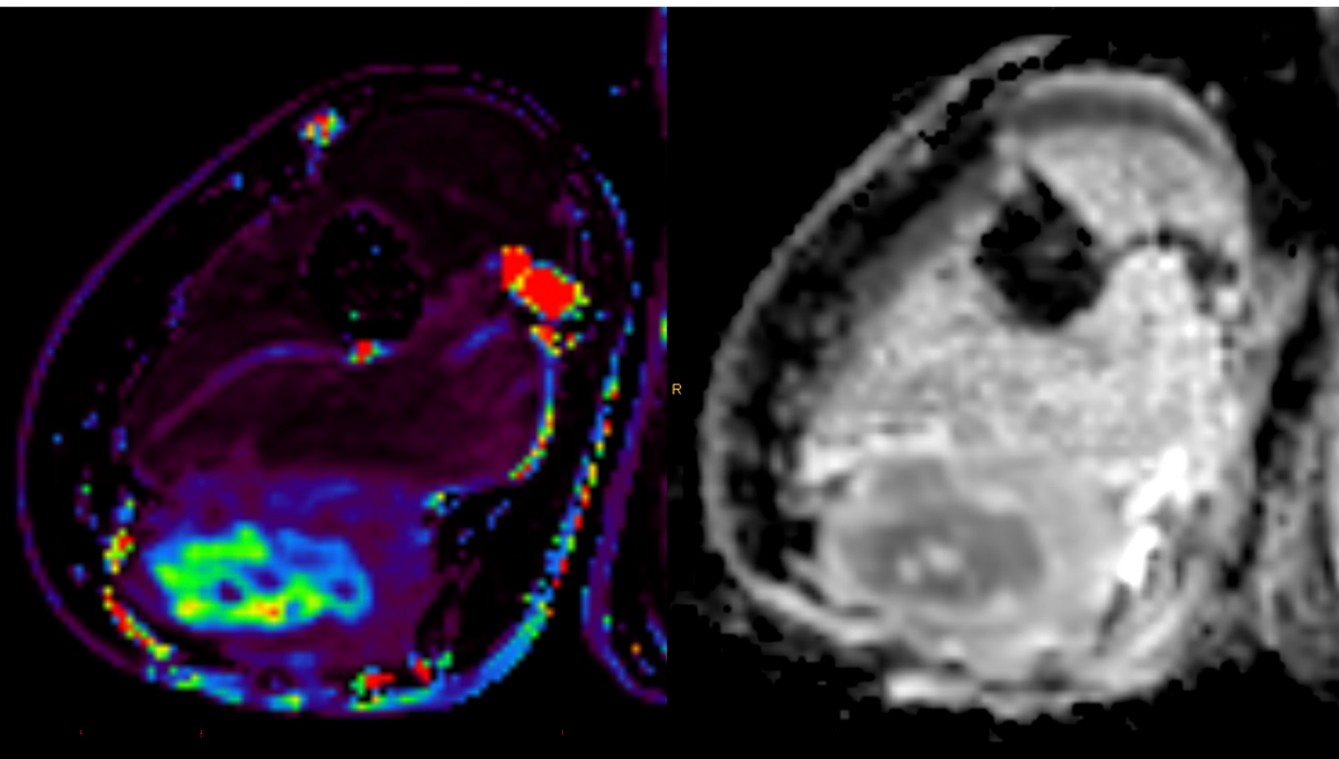

**Fig 6.** In-vivo K$^{trans}$ (left) and ADC (right) mapping of specific ROIs for CNBs.

## Library preparation for tumor, leukocyte, cfDNA samples, and NGS

**Exome sequencing.** Whole exome-sequencing (WES) is performed from native tissue DNA by CeGaT (Tübingen, Germany). Sequencing libraries are created according to the manufacturer's instructions using the Twist Human Core Exome Kit as well as the RefSeq and Mitochondrial Panel (Twist Bioscience, San Francisco, CA, USA). NovaSeq 6000 (Illumina Inc., San Diego, USA) is used for sequencing with read lengths of 2 x 100 bp. Output is 15 Gb per sample delivered in FAST-Q format files.

**Bioinformatics for exome sequencing.** We used a modified version of the MIRACUM pipe to identify the patient's genetic alterations based on whole-exome sequencing (WES) [46]. In brief: We used FASTQC for quality control [47] and processed the raw reads with Trimmomatic to remove Illumina adaptors and trailing bases with low-quality [48]. The trimmed reads were aligned to the hg19 reference genome [49] with Burrow-Wheeler Aligner (BWA)-MEM [50] and converted into a sorted bam file with samtools [51]. We used Varscan [52] to identify variants specific to tumor and normal tissue-kept variants with at least four variant-specific reads. Annovar [53] was used to annotate the results. To limit our dataset to likely oncogenic mutations, we used only rare mutations (< 0.1% global allele frequency across populations on GnomAD [54]). Subsequently, we generated a patient-specific panel targeting the genomic coordinates of the 30–90 tumor-specific variants with the highest variant allele frequency (VAF).

**Subtype-specific panel sequencing for myxoid liposarcomas and synovial sarcomas [33, 34].** DNA from tumor tissue (native or FFPE) and matched controls from whole blood samples are prepared for sequencing by generating libraries using NEB kit NEBNext Ultra II FS DNA Library Prep Kit for Illumina, (New England Biolabs, Ipswich, USA). DNA is fragmented to approximately 150 bp.

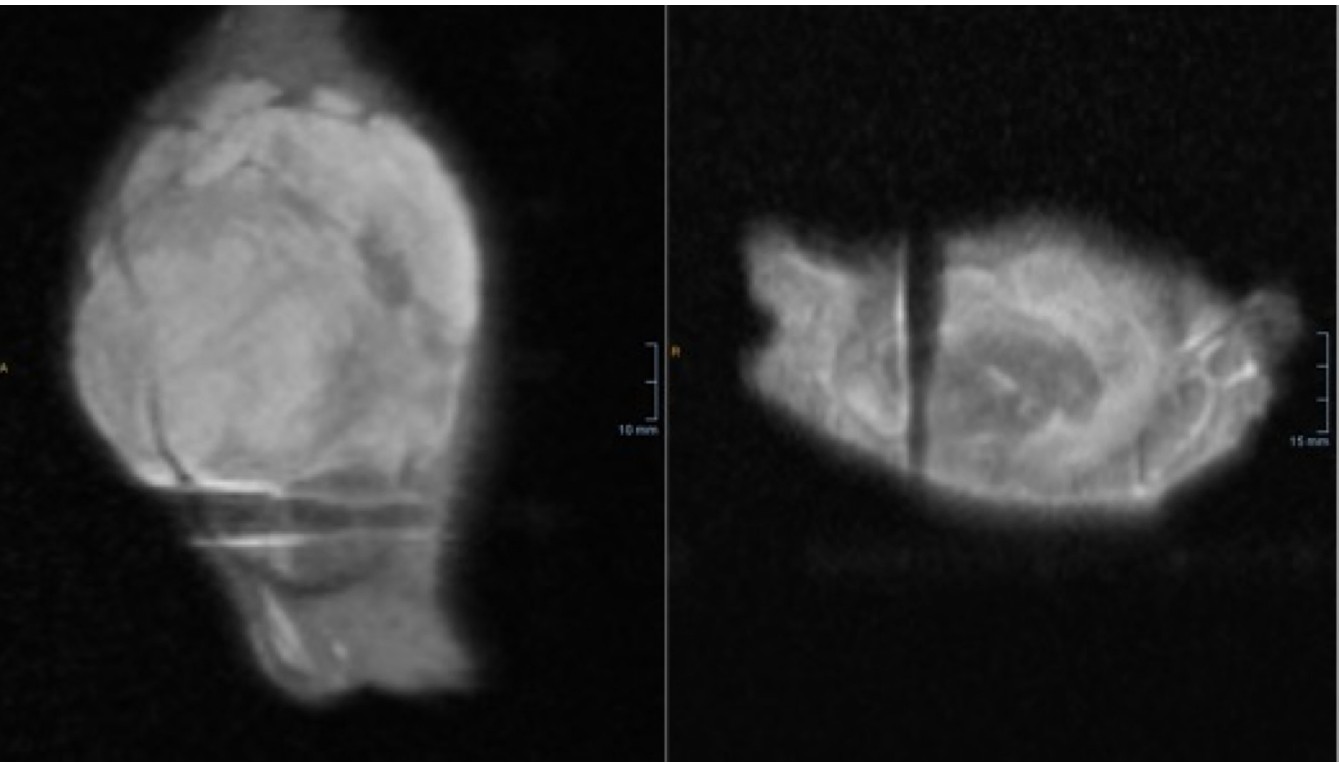

**Fig 7. CNB needle positioning is confirmed during an ex-vivo MRI scan of the native tumor specimen.**

Subtype-specific IDT xGen Panels comprising biotin probes are used to hybridize target regions. Following hybridization, target regions are pulled down with streptavidin-coated magnetic beads. The protocol will be performed according to the manufacturer's instructions, xGen hybridization capture of DNA libraries kit (Integrated DNA Technologies, Coralville, USA).

**Library preparation of cfDNA.** According to the manufacturer's manual, libraries are generated from cfDNA with the Takara SMARter Thruplex Tagseq Kit, 48S (Takara Bio In., Kusatsu, Shiga, Japan).

IDT xGen Panels comprising biotin probes are used to hybridize target regions. An individual tumor-specific lockdown panel is designed to encompass breakpoints and point mutations identified by panel and exome sequencing [33, 34]. Double target enrichment is performed with custom x-gen lockdown pool probes from the tumor-specific lockdown panel (Integrated DNA Technologies).

**Library quantification and sequencing.** The lengths of captured libraries are then measured by Tape Station with Agilent D500 screen tapes (Agilent, Santa Clara, USA). For accurate DNA concentration, libraries are quantified with the qPCR LightCycler 480 System (Roche), using the NEBNext Library Quant Kit for Illumina. The desired amount of these libraries is then calculated and sequenced using a Miseq system with paired-end reads, Miseq V2 300 cycle (Illumina Inc., San Diego, USA).

**Bioinformatics of cfDNA analysis.** The above-described sequencing data was then analyzed for structural variants (SVs) in patients' plasma samples. Consensus assembly, unique molecular identifier (UMI)-deduplication, and sequence alignment were performed with MAGERI computational pipeline [55]. The resulting Sequence Alignment Map (SAM) files were converted, sorted, and indexed with samtools [51]. With error-free alignment to the patient-specific breakpoint sequences, the presence of ctDNA was inferred.

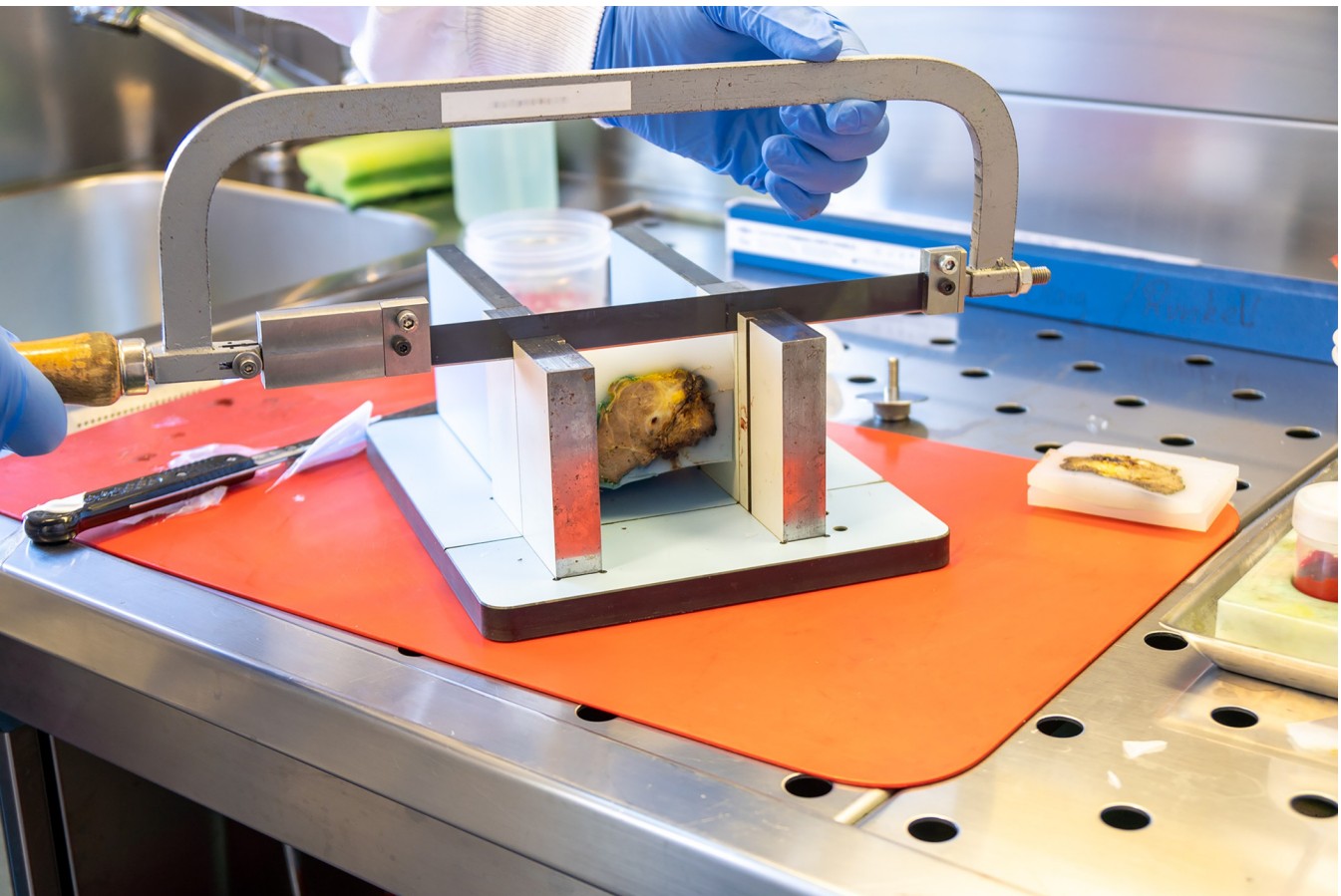

**Fig 8. Cutting of tumor slices for further histopathological analysis.**

Analysis of single nucleotide polymorphisms (SNPs) was performed using fgbio [56] (https://github.com/fulcrumgenomics/fgbio#license). Therefore, UMIs were extracted from a previously generated unmapped bam file. The extracted bam-file was aligned with BWA-MEM [50] to human reference genome GRCh38 and merged with the already mentioned unmapped bam-file. Afterward, reads were filtered with samtools. The reads were then grouped by UMI families and consensus reads were called using fgbio's "CallMolecularConsensusReads". The unmapped consensus reads were once again aligned to the human reference genome GRCh38 using BWA-MEM and indexed with samtools. The files generated hereby were then manually assessed for the previously identified mutations of the corresponding tumor tissue using the Integrative Genomics Viewer [57]. Mutations were scored as such if there were at least two alternating bases.

**Ethics, consent, and trial registry.**   The clinical study and establishment of the sarcoma Liquid-Biobank is in accordance with the Declaration of Helsinki and approved by the Ethics Committee of the Albert-Ludwig-University of Freiburg, Germany (study number: **EK 21–1735**). Signed and informed written consent is obtained from all participants before inclusion, allowing analysis of tumor tissue, blood samples, imaging, and clinical data.

This prospective study is registered in the German Clinical Trials Register (Deutsches Register Klinischer Studien DRKS, study number: **DRKS00027479**).

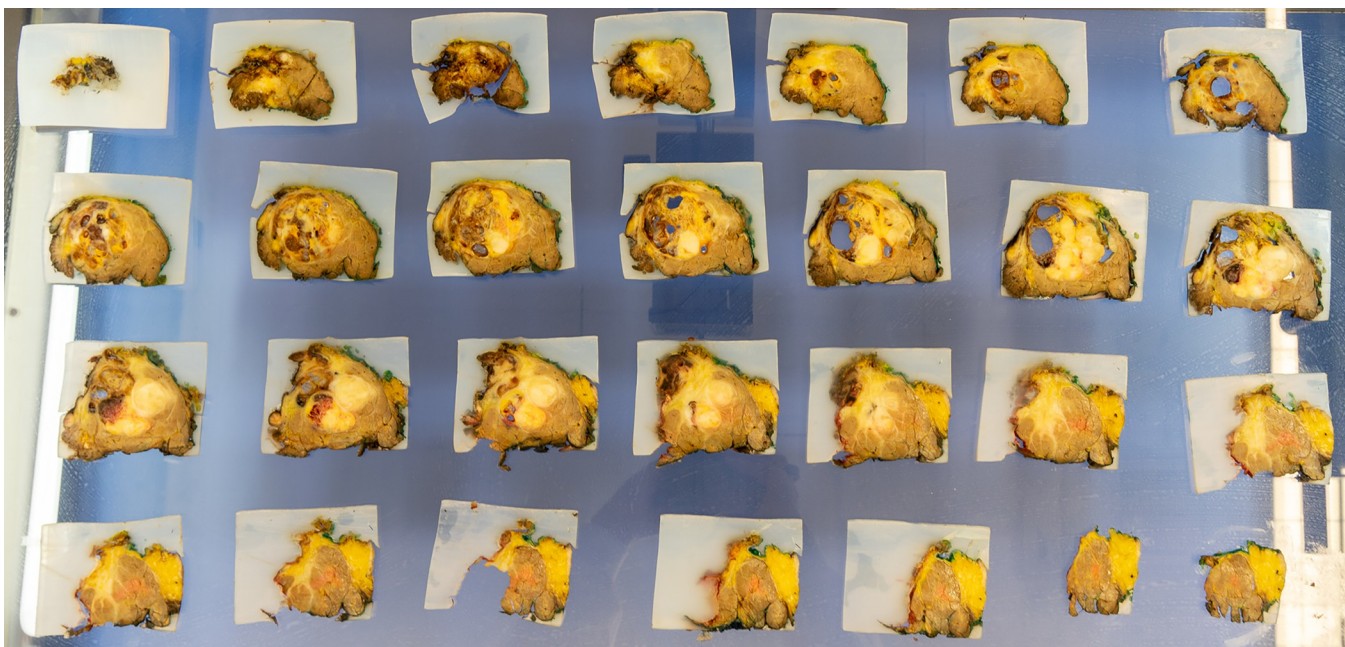

**Fig 9. Complete tumor specimen cut into 4mm thick slices.**

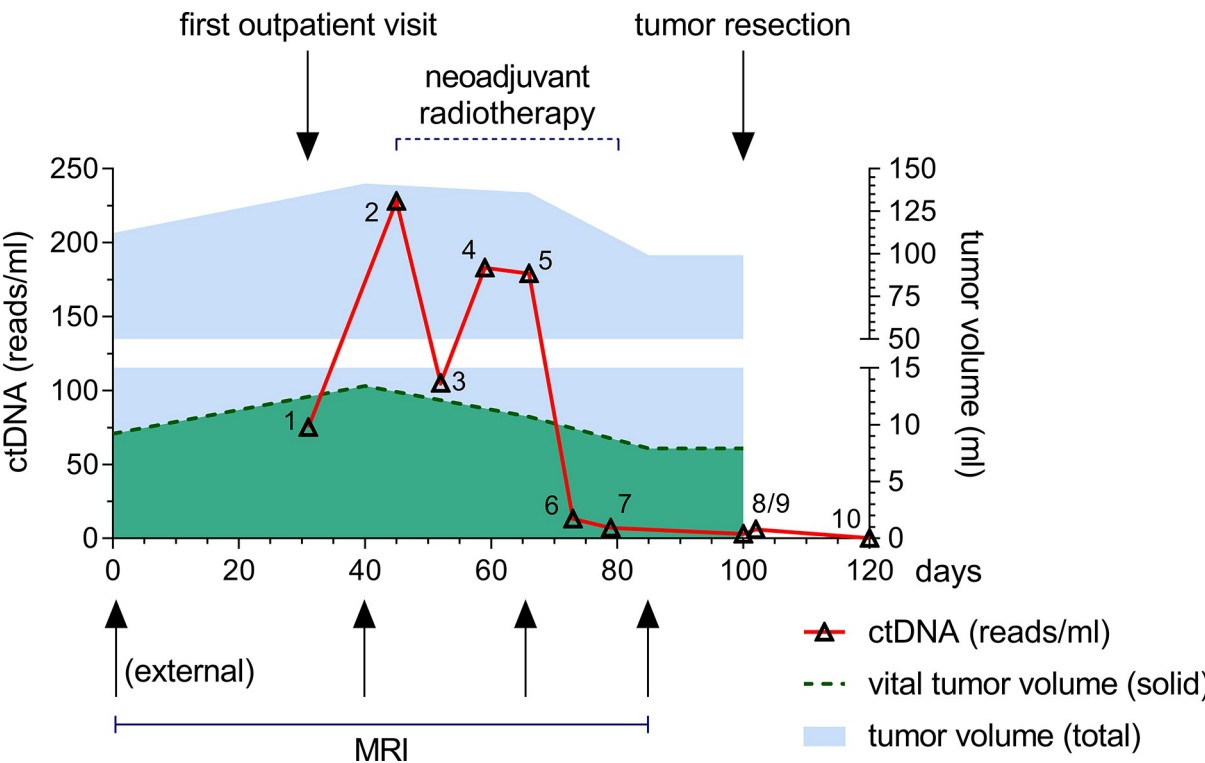

**Fig 10. Clinical course of patient 1.** Gross and vital tumor volumes and ctDNA levels before, during, and after NRT followed by complete resection of the tumor (light-blue area represents gross tumor volume; green area represents solid, vital tumor volume; the red line represents the quantity of ctDNA reads per ml peripheral blood; NRT was performed from timepoint 2–7; mpMRI were performed prior, during and after NRT; resection of the tumor occurred before timepoint 9). Three and six months after resection, no recurrence was identified in the follow-up MRI examination.

## Preliminary results

Exemplary, we present the clinical course of a pilot patient's ctDNA levels and tumor volumes before, during, and after NRT (Fig 10). Patient 1 presented with swelling in his right thigh and an external MRI as an outpatient in 2021 (day 0). Gross tumor volume at day 0 was 112 ml (light-blue area) with a solid, vital tumor volume of 9 ml (green area). CNBs were taken and analyzed that day. Histological diagnosis of a G2 Myxofibrosarcoma (MFS) was made by the Institute of Surgical Pathology, Medical Center—University of Freiburg, Faculty of Medicine, Freiburg, Germany. The patient was then presented to the interdisciplinary tumor board of the Comprehensive Cancer Center Freiburg two weeks later. Neoadjuvant radiation therapy was proposed and after study-specific and FREEZE-biobank informed consent, the patient was included in the prospective study on his second out-patient appointment one month after his initial visit. A liquid biopsy blood sample was drawn that day and it contained 79 ctDNA reads/ml (timepoint 1, red line). Patient 1 also received an mpMRI according to the study protocol. The following week, NRT was scheduled with the patient. Just before the first day of radiation, another liquid biopsy was taken (time point 2, red line). A drastic increase in ctDNA levels was observed between the time of diagnosis (79 reads/ml) and the start of neoadjuvant therapy (241 reads/ml). In addition, an increase in gross tumor volume (141 ml, light-blue area) and solid, vital tumor volume (13 ml, green area) was observed before the start of NRT. After one cycle of radiation therapy (5 x 2 Gy), another liquid biopsy was taken, and ctDNA levels were halved (112 reads/ml, timepoint 3, red line). After another cycle, ctDNA levels rose again to 187 reads/ml and fell slightly to 186 reads/ml after the third cycle of NRT (timepoint 4 and 5 respectively, red line). Before the start of the next cycle, another mpMRI was performed. Gross tumor volume decreased insignificantly to 136 ml (blue area) as well as solid tumor tissue (11 ml, green area) at day 68. It is interesting to note that during this time the tumor underwent considerable and quantitative changes. The initial solid, vital tumor tissue (high $K^{trans}$ and low ADC) decreased in size; however, solid, vital tumor tissue started growing in a different location inside the heterogenous MFS. Levels of ctDNA dropped significantly to 15 reads/ml after the fourth radiation cycle and continued to fall to 7 reads/ml after a total of 50 Gy radiation in five cycles (time points 6 and 7 respectively). Just before surgical resection, another mpMRI was performed. Gross tumor volume (light-blue area) fell just below 100 ml and the solid, vital tissue (green area) decreased to 8ml. Again, substantial morphological changes were observed in the mpMRI imaging data. Overall, solid, vital tumor tissue decreased in size, whereas necrotic, and myxoid tissue increased in volume. Only 3 ctDNA reads/ml were measured the day before the operation (timepoint 8). After complete tumor resection, ROI-based radio-pathological correlation analyses of the post-therapeutic specimen revealed increased values on the $K^{trans}$-map in vital tumor parts, characterized by cells positive for MIB1- and ALDH1A1, compared to necrotic tumor parts. The mean blood flow-related parameter $fD^*$ from the IVIM sequence was on average lower for vital tumor tissue for this patient. In line with mpMRI analyses, a histopathological assessment confirmed that vital tumor areas made up ~10% of the gross tumor volume. Pathologists diagnosed a grade 4 tumor regression analogous to *Salzer and Kuntschik* with a necrosis rate of ~40% and ~50% myxoid tumor tissue. Another liquid biopsy was taken three weeks after the operation (day 120). No ctDNA was found in the peripheral blood. The patient is still in remission based on the latest follow-up 6 months post-operatively.

## Discussion

In the literature [24, 58, 59] and previous works by our group [28–34, 45], ctDNA is described as a potential biomarker for the prediction of STS, its recurrence, and minimal residual disease

after resection. However, it has not been shown if ctDNA can be used as a biomarker for therapy response and how ctDNA correlates with vital tumor mass during NRT. To assess therapy response, the histopathology of the resected specimen has to be compared to a baseline, which, up to date, is represented by CNBs taken before the start of therapy. Yet, CNBs are taken from an immediate vicinity inside the often sizeable STS. Certainly, those biopsies only reflect a tiny fraction of the morphology and genetic landscape of the heterogeneous tumors and in some cases are insufficient to determine the STS subtype or grading [3, 60]. With no sufficient baseline, it is difficult to examine neoadjuvant therapy response and to propose adequate therapy concepts [61]. Consequently, different methods for the evaluation of therapy response in STS are required to establish evidence-based therapy concepts for patients with different STS subtypes. The proposed prospective study shows a new way to determine STS therapy response via a combination of non-/ minimal-invasive methods.

Are liquid biopsies alone sufficient to evaluate therapy response? Certainly, it is difficult to judge the vitality of a tumor using a single biomarker, since cellular processes at molecular levels are highly sophisticated. Yet, liquid biopsies and the detection of ctDNA have proven to be strong diagnostic tools to identify tumors of various kinds [62–65]. Thorough research has been performed in recent years to evaluate whether ctDNA could be used as a marker to monitor the clinical course of therapy [66]. For colorectal cancer, ctDNA levels have been shown to drop significantly after surgery [67, 68]. Another study from 2014 showed, that ctDNA levels for various cancers correlated with tumor stage or grading [69]. Our department has shown that ctDNA levels can be used to monitor the clinical course for various STS [29–34]. Therefore, we can assume that high levels of ctDNA in the peripheral blood signal high tumor burden and absence indicates full remission after e.g. surgery. However, how ctDNA levels evolve during neoadjuvant therapy could not be shown. The question remains whether ctDNA as a single biomarker directly correlates with tumor vitality. Does a higher rate of necrosis/apoptosis, which leads to high amounts of ctDNA being released into the peripheral bloodstream, indicate a higher rate of tumor growth and remodeling? During tumor growth, is ctDNA continuously released into the peripheral blood, or are certain triggers involved? As long as these questions remain unanswered ctDNA levels alone are insufficient to precisely correlate with tumor vitality, and thus therapy response.

Are radiological and pathological findings alone sufficient to monitor therapy response? Little evidence exists on how different STS subtypes respond to neoadjuvant therapy. A French study that measured gross tumor volume before and after NRT concluded, that Myxoid Liposarcomas (MLS) and Synovial Sarcomas (SS) are sensitive to radiation therapy due to a volume reduction, and Myxofibrosarcomas (MFS) do not respond well to NRT due to relative tumor volume growth [70]. Their retrospective study was solely based on radiological and pathological findings and did not measure therapy response based on the relative changes in vital tumor volume. In our previous studies and this prospective study, we have observed a few cases of tumor pseudo-progression under NRT. Gross tumor volume increased during therapy, but the relative amount of necrosis and edema also increased. Combining these findings with ctDNA quantification and mpMRI analysis, we could show, that despite volumetric pseudo-progression, the tumor released less ctDNA into the peripheral bloodstream, and mpMRI-based vitality markers decreased. In the results shown above, the MFS did not respond to neoadjuvant therapy with a considerable reduction in tumor volume. Yet, ctDNA levels fell drastically after the fourth cycle, and the final specimen's necrosis rate was 40% with a grade 4 regression analogous to *Salzer and Kuntschik*. Therefore, based on the patient example presented above, we cannot share the same conclusions as the French study group concerning MFS when combining the radiological and pathological data sets with ctDNA quantification. Radiological

findings based solely on gross tumor volume or pathological findings on the resected specimen alone are most likely not sufficient to monitor STS response to radiation therapy.

The results shown above are highly interesting since the levels of ctDNA and the quantitative and morphological changes in the radiological findings are very dynamic. Only the understanding of both sets of information shows why e.g. ctDNA levels fell after one cycle and rose again just before drastically falling after the fourth cycle. Due to a mpMRI approach, metabolically active tumor regions can be displayed and quantified using DWI, IVIM, and DCE sequences. These radiological findings show how STS morphology changed during NRT. The initial vital tissue was sensitive to early radiation, yet another vital part appeared during the second and third cycles, possibly explaining the increase in ctDNA levels. The mpMRI after the fourth and fifth cycles of NRT showed a predominant therapy response with a small remaining vital tumor part, which is consistent with the observed decrease in ctDNA. Therefore, combining liquid biopsies with multiparametric radiological imaging and meticulous pathological analysis can be the next step in characterizing tumor tissue and precisely monitoring tumor response.

All patients included fulfill the inclusion criteria, undergo NRT, have liquid biopsies drawn at the beginning of each therapy cycle, and, receive mpMRI at pre-defined time intervals at the same MR scanner. Inter-patient variability in imaging data is therefore reduced to a minimum. All protocols are standardized and reproducible (neoadjuvant radiation, follow-up, pathological assessment, panel design, bioinformatics). In addition, all patients included in this prospective study receive WES to reduce the rate of false positives due to germline mutations. Tumor mutations are identified by comparing the tumor exome with the matched patient leukocyte control. In addition, native tumor DNA is extracted from MR-guided punch biopsies to accurately capture the diverse genomic landscape of the STS, thus increasing the sensitivity of patient-specific panels. This ensures that SNPs and SVs from vital tumor tissue are accurately identified. We optimized the panel design through (I) enrichment of ctDNA, (II) use of molecular barcodes to decrease the rate of false positives, and (III) double hybridization capture, thereby increasing the on-target rate [71]. Exome sequencing comes at high financial costs but allows for a patient-specific panel design and the inclusion of tumor-specific mutations, which drastically increases the sensitivity of ctDNA detection. With lower sequencing costs, patient-specific exome panels should become clinical routine in the near future.

The study design and methods described in this article aim to establish the foundation of patient- and tumor-specific therapy recommendations. Large interdisciplinary data sets can be obtained using the above-described non-/ minimal-invasive methods of liquid biopsies and mpMRI in combination with the extensive pathological assessment of the resected tumor. The fusion of these different pieces of information will help to characterize the tumor and monitor changes in morphology during the course of therapy. In addition, ctDNA, combined with mpMRI can be used as a biomarker to monitor tumor response during radio- or chemotherapy. In the case of non-responder, prolonged neoadjuvant therapy can be aborted to prevent an unnecessary increase in tumor burden or even metastatic spread. Furthermore, different (neo)adjuvant therapy concepts for various STS subtypes can be developed using the above-described protocol for an elective range of sarcoma patients.

In summary, we have shown in preceding proof-of-concept studies, that ctDNA is a reliable biomarker to indicate the presence and absence of the primary STS and its metastases. In addition, ctDNA quantification mirrors the clinical course of treatment. Hence, we have developed a prospective study protocol that considers previous studies' limitations and uses a highly sensitive and optimized NGS-based panel design for liquid biopsies. Combining this data with mpMRI and immunohistological findings of the resected tumor, large interdisciplinary data sets are generated to characterize the STS [45]. In addition, tumor response to NRT and

morphological changes in the course of treatment can be monitored. Routine application of these minimally invasive methods can help to timely identify and treat recurrences, monitor treatment response, and create patient-specific therapy concepts at an affordable cost.

## Supporting information

**S1 Dataset. Quantification of ctDNA.** Raw and final data with calculations of ctDNA levels before, during, and after NRT with regard to changes in tumor masses.
(XLSX)

## Acknowledgments

The authors thank Dr. Dietmar Pfeifer and his team for the use of the MiSeq system, technical assistance, and advice. Marie Follo and the team of the Lighthouse Core Facility for their assistance with qPCR. Prof. S. Laßmann and her team for use of the Fragment Analyzer and NGS facility.

The authors particularly thank Mr. Gerd Strohmeier for constructing and manufacturing the custom-built angle plate and cutting device.

Authors AR and MJ are part of the **Berta-Ottenstein-Programme for Clinician Scientists**, Faculty of Medicine, University of Freiburg.

## Author Contributions

**Conceptualization:** Alexander Runkel, David Braig, Jurij Kiefer, Fabian Bamberg, Steffen Ulrich Eisenhardt, Matthias Jung.

**Data curation:** Alexander Runkel, Balazs Bogner, Ute Lausch, Anika Boneberg, Zacharias Brugger, Jurij Kiefer, Hannah Fuellgraf, Konrad Kurowski, Jutta Scholber, Philipp Rovedo, Matthias Jung.

**Formal analysis:** Alexander Runkel, Balazs Bogner, Adrian Schmid, Ute Lausch, Zacharias Brugger, Jurij Kiefer, Thomas Pauli, Melanie Boerries, Hannah Fuellgraf, Peter Bronsert, Philipp Rovedo, Steffen Ulrich Eisenhardt, Matthias Jung.

**Funding acquisition:** Alexander Runkel, Steffen Ulrich Eisenhardt.

**Investigation:** Alexander Runkel, Anika Boneberg, Steffen Ulrich Eisenhardt, Matthias Jung.

**Methodology:** Alexander Runkel, David Braig, Ute Lausch, Jurij Kiefer, Hannah Fuellgraf, Peter Bronsert, Jutta Scholber, Anca-Ligia Grosu, Fabian Bamberg, Steffen Ulrich Eisenhardt, Matthias Jung.

**Project administration:** Alexander Runkel, David Braig, Steffen Ulrich Eisenhardt, Matthias Jung.

**Resources:** Alexander Runkel, Ute Lausch, Anja Eisenhardt, Jurij Kiefer, Melanie Boerries, Konrad Kurowski, Peter Bronsert, Jutta Scholber, Fabian Bamberg, Steffen Ulrich Eisenhardt, Matthias Jung.

**Software:** Alexander Runkel, Balazs Bogner, Adrian Schmid, Thomas Pauli, Philipp Rovedo, Matthias Jung.

**Supervision:** Alexander Runkel, David Braig, Zacharias Brugger, Jurij Kiefer, Peter Bronsert, Jutta Scholber, Anca-Ligia Grosu, Philipp Rovedo, Fabian Bamberg, Steffen Ulrich Eisenhardt, Matthias Jung.

**Validation:** Alexander Runkel, David Braig, Balazs Bogner, Adrian Schmid, Ute Lausch, Anika Boneberg, Zacharias Brugger, Jurij Kiefer, Thomas Pauli, Melanie Boerries, Hannah Fuellgraf, Konrad Kurowski, Peter Bronsert, Anca-Ligia Grosu, Fabian Bamberg, Steffen Ulrich Eisenhardt, Matthias Jung.

**Visualization:** Alexander Runkel, Balazs Bogner, Adrian Schmid, Thomas Pauli, Hannah Fuellgraf, Philipp Rovedo, Matthias Jung.

**Writing – original draft:** Alexander Runkel.

**Writing – review & editing:** Alexander Runkel, David Braig, Balazs Bogner, Adrian Schmid, Ute Lausch, Anika Boneberg, Zacharias Brugger, Anja Eisenhardt, Jurij Kiefer, Thomas Pauli, Melanie Boerries, Hannah Fuellgraf, Konrad Kurowski, Peter Bronsert, Anca-Ligia Grosu, Philipp Rovedo, Fabian Bamberg, Steffen Ulrich Eisenhardt, Matthias Jung.

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
