## [Decision Letter · Decision Letter 0]

6 Feb 2023

PONE-D-22-31540

Non-invasive monitoring of neoadjuvant radiation therapy response in soft tissue sarcomas by multiparametric MRI and quantification of circulating tumor DNA -- a study protocol

PLOS ONE

Dear Dr. Runkel,

Thank you for submitting your manuscript to PLOS ONE. After careful consideration, we feel that it has merit but does not fully meet PLOS ONE’s publication criteria as it currently stands. Therefore, we invite you to submit a revised version of the manuscript that addresses the points raised during the review process.

We look forward to receiving your revised manuscript.

Kind regards,

Alvaro Galli

Academic Editor

PLOS ONE

Journal Requirements:

3. lease include your full ethics statement in the ‘Methods’ section of your manuscript file. In your statement, please include the full name of the IRB or ethics committee who approved or waived your study, as well as whether or not you obtained informed written or verbal consent. If consent was waived for your study, please include this information in your statement as well.

"This project is funded by Freiburg’s Forschungskommission (FoKo).

Authors AR and MJ are funded by the Berta-Ottenstein-Programme for Clinician Scientists, Faculty of Medicine, University of Freiburg.

"

"Author AR received funding from Freiburg’s Forschungskommission (FoKo).

Authors AR and MJ are funded by the Berta-Ottenstein-Programme for Clinician Scientists, Faculty of Medicine, University of Freiburg.

The funders had and will not have a role in study design, data collection and analysis, decision to publish, or preparation of the manuscript"

Reviewers' comments:

Reviewer's Responses to Questions

**Comments to the Author**

1. Does the manuscript provide a valid rationale for the proposed study, with clearly identified and justified research questions?

Reviewer #1: Yes

Reviewer #2: Yes

Reviewer #3: Partly

2. Is the protocol technically sound and planned in a manner that will lead to a meaningful outcome and allow testing the stated hypotheses?

Reviewer #1: Yes

Reviewer #2: Yes

Reviewer #3: No

3. Is the methodology feasible and described in sufficient detail to allow the work to be replicable?

Reviewer #1: No

Reviewer #2: Yes

Reviewer #3: No

4. Have the authors described where all data underlying the findings will be made available when the study is complete?

Reviewer #1: Yes

Reviewer #2: No

Reviewer #3: Yes

5. Is the manuscript presented in an intelligible fashion and written in standard English?

Reviewer #1: Yes

Reviewer #2: Yes

Reviewer #3: Yes

6. Review Comments to the Author

You may also provide optional suggestions and comments to authors that they might find helpful in planning their study.

Reviewer #1: Runkel et al. planed to conduct a prospective study to assess the decision of neoadjuvant radiation therapy (NRT) in soft-tissue sarcomas (STS) patients using ctDNA and MRI data. The proposed study is clinically important because patients with NRT are up to two times more likely to develop wound healing disorders postoperatively. Therefore, it is important to evaluate which STS patient subpopulation and tumor subtype would benefit from NRT. There are a few minor points about the current manuscript, which are listed below.

1. The method section of the abstract should also include the information of patients that intend to be recruited.

2. I found two figure 1s in the manuscript, one on the lines 183-203, and the other at the end of the manuscript.

3. What are the proposed statistical methods for analyzing the combination of ctDNA and MRI results? What’s the plan for validating the results?

Reviewer #2: The current manuscript presents an innovative study protocol for monitoring soft tissue sarcomas (STS) using circulating tumor DNA (ctDNA) and multiparametric MRI. The hypothesis, however, is “that due to STS heterogeneity, subtypes respond differently to NRT”. Furthermore “it is important to evaluate which STS subtypes benefit from NRT.” I congratulate the authors on their goal dealing with one of the “hottest topics” in the field of STS.

Although the methods are clear, up-to-date and the authors have a lot of well published experience with ctDNA, I recommend to sharpen the hypothesis concerning the use of ctDNA:

We already know, that several STS subtypes respond differently to neoadjuvant radiotherapy – e.g. myxoid liposarcoma (good) in comparison to myxofibrosarcoma (see your own discussion). Using ctDNA to monitor a neoadjuvant approach is an excellent idea and has been already introduced for Ewing Sarcoma (Seidel MG et al, Front Pediatr 2022). In this study the authors reported quite large intra-individual differences in copy number levels, something reported by Krumbholz M et al (Clin Cancer Res 202) on a larger level - Pretreatment ctDNA copy numbers correlated with EFS & OS.

So – if the authors want to monitor any neoadjuvant local treatment (RTX) they would only be able to include localized, non-metastastic disease, something I did not find in the exclusion criteria – which is only mentioned in the abstract and the introduction. Then, however, the number possible to include will shrink, possibly making it necessary to include another center? However, maybe the individual copy number levels will help us to decide which patients might profit from (neo-)adjuvant chemotherapy – or even (neo-)adjuvant radiotherapy?

Minor remarks:

A more extensive “Data Availability Statement” for the data collected in the study is recommended.

Page 4, line 71 – “At the time of diagnosis, local tumor growth or metastatic spread has already occurred …” - “local tumor growth” is unspecific, I would add “over 5 cm” – as this cut off has been published to have a negative influence on OS

Page 4, line 76 – metastastes can only “reoccur” when they have been treated before, too (e.g. resected) – and not after “primary tumor resection”

Page 5, line 100 – first time (besides the abstract), that you use the abbreviation ctDNA – please write in full. The same applies for NRT (page 4)

Reviewer #3: Comments to the authors

General comments

In this study, each limitation of liquid biopsy, multi-parameter MRI, and histopathology of the resected specimen for the assessment of therapy response to STS was described, and a protocol combining those methods has been proposed. This protocol allows the prevention of unnecessary treatment by monitoring of therapy response, and proposes patient-specific therapy concept. This study is meaningful because it will benefit many patients.

However, there are some questions. Comments on this study are provided below.

Major comments

Correlation between ctDNA and each evaluated value

1. ctDNA is important role in this study. ctDNA and tumor volume during the NRT were measured to evaluate the therapy response, but how is the correlation between changes of the ctDNA and the volume? It seems likely that ctDNA would correlate with the vital tumor volume, but the vital tumor volume did not seem to have changed much in the results of this study.

2. Although the evaluated values by MRI, such as ADC, D*, f, Ktrans, and etc., in the proposed protocol, the correlation between ctDNA and these value should be mentioned.

3. How is the correlation between amount of change of ctDNA during the NRT and the final specimen’s necrosis rate? If there is a strong correlation, it seems that ctDNA during the NRT reflects the therapy response.

Tumor volume measurements

4. In this study the tumor volume was measured during the NRT, what pulse sequence was used?

5. Cross tumor volume and vital tumor volume were measured in results. Vital tumor was defined as a high Ktrans, and low ADC, please indicate those thresholds.

Other

6. In the results and beyond section, there is no mention of the IVIM. Is IVIM necessary in the proposed protocol?

7. In this study, there is only one case. Is it possible to increase subjects? One case is insufficient to establish the validity of the proposed protocol.

Minor comments

Please standardize the format of abbreviations. There are characters that are not translations again after the abbreviation is defined. There are words for which abbreviations are not defined (e.g. SNPs). Please confirm.

7. PLOS authors have the option to publish the peer review history of their article (what does this mean?). If published, this will include your full peer review and any attached files.

Reviewer #1: No

Reviewer #2: No

Reviewer #3: No

---

## [Author Response · Author response to Decision Letter 0]

10 Mar 2023

Revision of manuscript PONE-D-22-31540

Non-invasive monitoring of neoadjuvant radiation therapy response in soft tissue sarcomas by multiparametric MRI and quantification of circulating tumor DNA – a study protocol

Dear Dr. Galli,

Please reconsider our manuscript entitled “Non-invasive monitoring of neoadjuvant radiation therapy response in soft tissue sarcomas by multiparametric MRI and quantification of circulating tumor DNA – a study protocol” for publication in PLOS One. We highly appreciate the reviewers’ comments that helped us to substantially improve the manuscript. 

As suggested by the reviewers, we have addressed all issues raised regarding the statistical analysis and the correlation between ctDNA levels and multiparametric imaging data and added new tables and calculations to this rebuttal letter. Figure 9 has been updated in higher resolution and updated on the y-axis. The data will be made available to you and is attached to the Supporting Information section. 

However, we suggest not including more patient data. We believe that the pilot data presented advocates the feasibility of our study and should not serve as a validation of the methods. Since STS patients who fit our inclusion criteria are rare, we would like to publish any consecutive patient data as part of a future article with PLOS One after the study completion.

In-depth point-by-point responses to the reviewers’ comments are adjoined.

Please find the revised manuscript with tracked changes, the updated manuscript, the updated Fig. 9, and a supporting dataset in the submitted files. Changes are marked and underlined in red; omissions are crossed out. Please consider our responses to your comments below.

Thank you again for your consideration.

Best regards.

Alexander Runkel

 

Response to editor

Response: The requirements have been studied carefully and file naming has been adjusted. 

Response: Written informed consent was obtained from all participants prior to inclusion. The missing information has been added in line 392 now moved to the methods section.

“Signed and informed written consent is obtained from all participants before inclusion, allowing analysis of tumor tissue, blood samples, imaging, and clinical data.”

Response: The entire “Ethics, consent, and trial registry” paragraph has been moved to the methods section (lines 389-395). Additional information about written informed consent is now included.

“The clinical study and establishment of the sarcoma Liquid-Biobank is in accordance with the Declaration of Helsinki and approved by the Ethics Committee of the Albert-Ludwig-University of Freiburg, Germany (study number: EK 21-1735). Signed and informed written consent is obtained from all participants before inclusion, allowing analysis of tumor tissue, blood samples, imaging, and clinical data.

This prospective study is registered in the German Clinical Trials Register (Deutsches Register klinischer Studien DRKS, study number: DRKS00027479).”

Response: Please refer to the response to the next comment below. A grant number has been added.

"This project is funded by Freiburg’s Forschungskommission (FoKo).

Authors AR and MJ are funded by the Berta-Ottenstein-Programme for Clinician Scientists, Faculty of Medicine, University of Freiburg."

"Author AR received funding from Freiburg’s Forschungskommission (FoKo).

Authors AR and MJ are funded by the Berta-Ottenstein-Programme for Clinician Scientists, Faculty of Medicine, University of Freiburg.

The funders had and will not have a role in study design, data collection and analysis, decision to publish, or preparation of the manuscript".

Response: Thank you for pointing out this inconsistency. All funding related text is removed from the manuscript. The updated funding statement should read as follows:

 "Author AR received funding from Forschungskommission Freiburg (Grant number: 3095120035).

Authors AR and MJ are part of the Berta-Ottenstein-Programme for Clinician Scientists, Faculty of Medicine, University of Freiburg.

The funders had and will not have a role in study design, data collection and analysis, decision to publish, or preparation of the manuscript".

Thank you for updating the Funding Statement.

Response: The study’s minimal underlying data set is now attached as a Supporting Information file (S1 Dataset.xlsx).

Thank you for updating the Data Availability Statement.

 

Response to Reviewers

Reviewer #1

Runkel et al. planed to conduct a prospective study to assess the decision of neoadjuvant radiation therapy (NRT) in soft-tissue sarcomas (STS) patients using ctDNA and MRI data. The proposed study is clinically important because patients with NRT are up to two times more likely to develop wound healing disorders postoperatively. Therefore, it is important to evaluate which STS patient subpopulation and tumor subtype would benefit from NRT. There are a few minor points about the current manuscript, which are listed below.

Response:

Dear Reviewer 1, thank you for your comments. As you mentioned, it is important to investigate from a molecular to a macroscopic level, which STS subtypes benefit from NRT to avoid e.g. postoperative complications. I would like to answer your points below:

1. The method section of the abstract should also include the information of patients that intend to be recruited.

Response: Information about the recruitment of patients is now included in line 44:

 “Patients with localized high-grade STS, who qualify for NRT, are included in this study.”

2. I found two figure 1s in the manuscript, one on the lines 183-203, and the other at the end of the manuscript.

Response: Fig. 1 refers to the timeline of the prospective study protocol attached at the end of the manuscript. The Flowchart in Section “Patient diagnostics and therapy steps” (lines 183-203 in the former manuscript) is not considered to be a figure, but rather an integral part of the manuscript itself. In order to avoid confusion, the Fig.1 title has been moved to line 175.

3. What are the proposed statistical methods for analyzing the combination of ctDNA and MRI results? What’s the plan for validating the results?

Response: We included an entire section in the study protocol, outlining the statistical models and necessary sample size for this study. Please refer to “Statistical analysis and sample size calculation” (lines 284-293):

“During study planning, we made a sample size calculation in collaboration with the Institute of Medical Biometry and Statistics, IMBI, Freiburg, Germany. We plan to correlate mpMRI parameters (DWI (i.e. ADC), IVIM (i.e. f, D*, and fD*), and DCE (i.e. Ktrans, ve, kep, vp) ) with ctDNA levels and immunohistology (cell density proliferation index and ALDH1A1 radiation resistance and MIB1 proliferation marker). We will analyze at least five ROIs (fixed effects) in each STS patient (random effect), which makes our data nested. Thus, a mixed effects model for nested data is necessary for analyzing associations of quantitative mpMRI parameters with ctDNA levels and (immuno)histology. For modeling with up to five predictors – in our case the quantitative mpMRI sequences – a sample size of 25 patients is necessary for a reliable model.”

In addition, the correlation between mpMRI data and histology has also been outlined and described in section “Statistical clustering and mpMRI – Histology – ctDNA correlation” lines 296-310:

“The quantitative map values of the DWI (i.e. ADC), IVIM (i.e. f, D*, and fD*), and DCE (i.e. Ktrans, ve, kep, vp) sequence are extracted for each voxel within the tumor ROI. Parametric map values within the tumor ROI are equally weighted for subsequent classification into four groups, based on histological analysis. We perform K-means clustering (Python machine learning library scikit-learn (1)) to split the voxels of the tumor ROI into a set of four groups. The classification and voxel position identified by clustering are mapped onto the T2 sequence by color coding. Clustering is performed using only parametric mpMRI-values. Spatial information is not used as an additional input, but only to identify the clusters on the T2 image. Four representative ROIs - one per cluster – are defined in the co-registered histology and mpMRI section. One-way ANOVA with post hoc pairwise t-tests (Bonferroni-Holm adjusted) is performed to compare differences in mpMRI parameters.”

These statistical methods are explained in depth in our other manuscript published in the current issue of Theranostics (2).

The validation of results will follow by inclusion of more patients in this prospective study. With around 25 data sets our statistical model will be reliable enough to validate or reject our hypothesis.

 

Reviewer #2

The current manuscript presents an innovative study protocol for monitoring soft tissue sarcomas (STS) using circulating tumor DNA (ctDNA) and multiparametric MRI. The hypothesis, however, is “that due to STS heterogeneity, subtypes respond differently to NRT”. Furthermore “it is important to evaluate which STS subtypes benefit from NRT.” I congratulate the authors on their goal dealing with one of the “hottest topics” in the field of STS.

Although the methods are clear, up-to-date and the authors have a lot of well published experience with ctDNA, I recommend to sharpen the hypothesis concerning the use of ctDNA:

We already know, that several STS subtypes respond differently to neoadjuvant radiotherapy – e.g. myxoid liposarcoma (good) in comparison to myxofibrosarcoma (see your own discussion). Using ctDNA to monitor a neoadjuvant approach is an excellent idea and has been already introduced for Ewing Sarcoma (Seidel MG et al, Front Pediatr 2022). In this study the authors reported quite large intra-individual differences in copy number levels, something reported by Krumbholz M et al (Clin Cancer Res 202) on a larger level - Pretreatment ctDNA copy numbers correlated with EFS & OS.

So – if the authors want to monitor any neoadjuvant local treatment (RTX) they would only be able to include localized, non-metastastic disease, something I did not find in the exclusion criteria – which is only mentioned in the abstract and the introduction. Then, however, the number possible to include will shrink, possibly making it necessary to include another center? However, maybe the individual copy number levels will help us to decide which patients might profit from (neo-)adjuvant chemotherapy – or even (neo-)adjuvant radiotherapy?

Response:

Dear Reviewer 2, thank you for your detailed response and great insight into ctDNA copy numbers for Ewing sarcomas. We share your enthusiasm about this “hot topic” and wish to contribute to the knowledge about ctDNA response during NRT. We agree, that ctDNA copy number variations are highly interesting with regard to overall and event-free survival. We also perform Methylation-Analyses (850k EPIC Array) for STS with copy number variation profiling, which will be the focus of a separate study. Over a long-term follow-up period, such correlations will also be analyzed using the study protocol described here. With such data, pretreatment ctDNA copy numbers can be implemented into the therapy decision algorithm. The hypothesis has been modified as requested (lines 122-127):

“We hypothesize, that due to STS heterogeneity, subtypes respond differently to NRT, as shown in absolute ctDNA copy numbers and relative changes during therapy. In some cases, early tumor resection might lower surgical complications and improve survival rates. Pretreatment copy numbers of ctDNA can be used as a predictive and prognostic biomarker for Ewing sarcomas ((3)). We hypothesize that pretreatment ctDNA levels might correlate with overall and event-free survival for STS patients.”

In our prospective study, we intend to only monitor patients who receive NRT. According to the current S3 guidelines, only high-grade STS qualify for such a treatment. The exclusion criteria have been adjusted accordingly in line 165:

“Exclusion criteria are low grade STS, dementia, anemia, age < 16, pregnancy, or the inability to consent.”

Given these inclusion criteria, the number of patients that can be included in this study is limited. However, over the past 12 months, we treated >10 STS patients that qualified for inclusion in this study. Over three years we shall therefore acquire >25 study patients, which is the necessary amount to perform statistical analysis according to our sample size calculation (refer to added section “Statistical analysis” lines 284-293). To validate our results, a multicenter approach is necessary, due to the rarity of STS in general.

I would like to answer your remaining remarks below:

1. A more extensive “Data Availability Statement” for the data collected in the study is recommended.

Response: The “Data Availability Statement” has been revised and updated by the editor. A minimal underlying data set will be included in this revision of the manuscript.

2. Page 4, line 71 – “At the time of diagnosis, local tumor growth or metastatic spread has already occurred …” - “local tumor growth” is unspecific, I would add “over 5 cm” – as this cut off has been published to have a negative influence on OS

Response: The additional information has been added in line 77 as requested:

“At the time of diagnosis, local tumor growth (>5cm) or metastatic spread has often already occurred, leading to a significant reduction in patient quality of life and overall survival (4).”

3. Page 4, line 76 – metastases can only “reoccur” when they have been treated before, too (e.g. resected) – and not after “primary tumor resection”

Response: Thank you for the notice. The typo has been corrected in line 81 and now reads “occur”.

4. Page 5, line 100 – first time (besides the abstract), that you use the abbreviation ctDNA – please write in full. The same applies to NRT (page 4)

Response: Thank you for pointing out these discrepancies. Abbreviations that have been used in the abstracts are now written in full in the main manuscript body. In addition, the format of abbreviations has been revised throughout the manuscript according to journal's requirements. Reviewer 3 also pointed out inconsistencies in the use of abbreviations.

 

Reviewer #3

In this study, each limitation of liquid biopsy, multi-parameter MRI, and histopathology of the resected specimen for the assessment of therapy response to STS was described, and a protocol combining those methods has been proposed. This protocol allows the prevention of unnecessary treatment by monitoring of therapy response and proposes patient-specific therapy concept. This study is meaningful because it will benefit many patients.

However, there are some questions. Comments on this study are provided below.

Response: Dear Reviewer #3, thank you for your response and your major comments, which allowed us to strengthen our manuscript. I would like to reply to the major comments below:

1. ctDNA is an important role in this study. ctDNA and tumor volume during the NRT were measured to evaluate the therapy response, but how is the correlation between changes of the ctDNA and the volume? It seems likely that ctDNA would correlate with the vital tumor volume, but the vital tumor volume did not seem to have changed much in the results of this study.

Response: We hypothesize, that ctDNA levels correlate with vital tumor volume. Vital tumor volume is measured via segmentation using the in-house nora medical imaging platform (https://www.nora-imaging.com). MRI images, which are acquired before, during, and after NRT give us three timepoints at which tumor volumes can be correlated with ctDNA levels. In our preliminary data set vital volume and ctDNA are shown below:

timepoint day vital tumor volume (ml) ctDNA (reads/ml)

2 40 13,4 228

5 66 10,7 179

8 85 7,9 7

timepoint days between measurements relative change in vital tumor volume relative change in ctDNA reads/ml

5 26 20,1% 21,5%

8 19 26,2% 96,1%

The table shows, that within the three weeks between the first mpMRI (day 40) and the second mpMRI (day 66) the relative change in vital tumor tissue was 20,1% vs. relative change in ctDNA levels 21,5% seem to correlate. However, the relative changes of vital tumor tissue and ctDNA levels between the second mpMRI (day 66) and the final mpMRI after NRT (day 85) are divergent (26,2% vs. 96,1%). Based on just two measurements our hypothesis cannot be substantiated. More data is needed to infer data trends.

In order to show the changes in vital tumor volume more clearly, our results graph (Fig. 9) has been modified by splitting the y-axis regarding tumor volume (ml). In addition, the resolution has been increased. The new graph (Fig 9.tif ) is uploaded with the revision of this manuscript.

2. Although the evaluated values by MRI, such as ADC, D*, f, Ktrans, and etc., in the proposed protocol, the correlation between ctDNA and these value should be mentioned.

Response: Thank you for pointing this out. We included an entire section in the study protocol, outlining the statistical models and necessary sample size for this study. Here the correlation between ctDNA and the mpMRI parameters is mentioned. Please refer to “Statistical clustering and mpMRI – Histology – ctDNA correlation” lines 296-310:

“The quantitative map values of the DWI (i.e. ADC), IVIM (i.e. f, D*, and fD*), and DCE (i.e. Ktrans, ve, kep, vp) sequence are extracted for each voxel within the tumor ROI. Parametric map values within the tumor ROI are equally weighted for subsequent classification into four groups, based on histological analysis. We perform K-means clustering (Python machine learning library scikit-learn (1)) to split the voxels of the tumor ROI into a set of four groups. The classification and voxel position identified by clustering are mapped onto the T2 sequence by color coding. Clustering is performed using only parametric mpMRI-values. Spatial information is not used as an additional input, but only to identify the clusters on the T2 image. Four representative ROIs - one per cluster – are defined in the co-registered histology and mpMRI section. One-way ANOVA with post hoc pairwise t-tests (Bonferroni-Holm adjusted) is performed to compare differences in mpMRI parameters.”

3. How is the correlation between amount of change of ctDNA during the NRT and the final specimen’s necrosis rate? If there is a strong correlation, it seems that ctDNA during the NRT reflects the therapy response.

Response: A correlation between ctDNA and the necrosis rate is highly interesting. However, there are two limitations which could weaken the informative value of such a correlation: 

1. We only have two time points where necrosis rate is measured by the pathologists: the CNB during the patient’s first out-patient visit; and the resected specimen after operation.

2. A histopathological assessment of the true necrosis rate of the tumor prior to therapy based on a few CNBs seems very difficult due to the size and heterogeneity of STS. CNBs could be taken in areas which an under- or overrepresentation of necrotic tissue. 

If we look at our data, the pathologists described a pre-therapeutic and post-NRT necrosis rate of 10% and 40% respectively, which results in a 4-fold increase. Levels of ctDNA were 79 reads/ml at the time of the CNB to 3 reads/ml at the time of operation, which is a 26-fold decrease. It seems as if necrosis rate and ctDNA are inversely correlated, however a single data point is insufficient to derive any data trends or mathematical relationship.

 

timepoint day necrosis rate (%) relative increase in necrosis rate (fold) ctDNA (reads/ml) relative decrease in ctDNA level (fold)

1 31 5 79 

8 100 40 8,0 3 26,3

4. In this study the tumor volume was measured during the NRT, what pulse sequence was used?

Response: Tumor volume is not measured directly during NRT, but calculated from MRI images, which are acquired before, during, and after NRT. Vital tumor volume is measured via segmentation using the in-house nora medical imaging platform (https://www.nora-imaging.com).

5. Cross tumor volume and vital tumor volume were measured in results. Vital tumor was defined as a high Ktrans, and low ADC, please indicate those thresholds.

Response: In the table below we listed the values for Ktrans, ADC and fD* cutoffs. However, the interpretation of these values should be done with caution, since they are based on the single pilot data set, and can vary during the analysis of further patient data.

 DCE (Ktrans) [1/min] DWI (ADC) [10^-3 mm^2/s] IVIM (fD*) [10^-3 mm^2/s]

Tumor Volume mean median threshold mean median threshold mean median threshold

Vital 229 201 180 1599 1582 1750 1,76 1,53 2,1

Total 90 65 2112 2010 2,5 1,31 

It is important to note that each individual threshold mentioned above is not a definite cutoff that separates vital from non-vital tissue. It must be emphasized that a definitive specification of cutoffs can only be provided after final evaluation of the prospective study presented here with a case number of approximately 25 patients. In addition, an isolated analysis of the individual parametric mappings should be avoided, as it has already been shown in previous studies that only a combination of the parametric maps can depict the complex biological behavior and tumor microenvironment of heterogeneous STS.

6. In the results and beyond section, there is no mention of the IVIM. Is IVIM necessary in the proposed protocol?

Response: Thank you for noticing. The blood flow-related parameter fD* from the IVIM sequence has been added in the results section. Yes, IVIM is an additional sequence necessary for our multiparametric approach of our proposed protocol. Previous studies highlight the importance of a multiparametric approach to identify differences in histological and biological characteristics of heterogeneous STS tumor parts (5,6).

7. In this study, there is only one case. Is it possible to increase subjects? One case is insufficient to establish the validity of the proposed protocol.

Response: We are aware, that a single patient’s data set is insufficient to infer statistical trends. However, the PLOS One guidelines for study protocols state that inclusion of data is optional and that pilot data should be included, “only if it is necessary to support the feasibility of the study or as a proof of principle.” Since STS patients who fit our inclusion criteria are rare, we would like to publish any consecutive patient data as part of a future article with PLOS One after the study completion. A sample size of 25 patients is necessary for a reliable statistical model that can be used for validation. We believe that the pilot data presented here advocates the feasibility of our study and should not serve as a validation of the methods.

I would like to respond to your minor comment below:

1. Please standardize the format of abbreviations. There are characters that are not translations again after the abbreviation is defined. There are words for which abbreviations are not defined (e.g. SNPs). Please confirm.

Response: A lack of consistency with regard to abbreviations has been mentioned by other reviewers as well. Therefore, unnecessary abbreviations have been omitted and abbreviations first defined in the abstract are now defined in the main manuscript body as well. Changes have been made throughout the entire manuscript and are highlighted using track changes. I hope this clarifies your concern.

 

Bibliography:

1. Abraham A, Pedregosa F, Eickenberg M, Gervais P, Mueller A, Kossaifi J, et al. Machine learning for neuroimaging with scikit-learn. Front Neuroinform [Internet]. 2014 Feb 21 [cited 2023 Feb 15];8(FEB). Available from: https://pubmed.ncbi.nlm.nih.gov/24600388/

2. Jung M, Bogner B, Diallo TD, Kim S, Arnold P, Füllgraf H, et al. Multiparametric magnetic resonance imaging for radiation therapy response monitoring in soft tissue sarcomas: a histology and MRI co-registration algorithm. Theranostics [Internet]. 2023 [cited 2023 Mar 7];13(5):1594–606. Available from: https://www.thno.org//creativecommons.org/licenses/by/4.0/

3. Krumbholz M, Eiblwieser J, Ranft A, Zierk J, Schmidkonz C, Stütz AM, et al. Quantification of Translocation-Specific ctDNA Provides an Integrating Parameter for Early Assessment of Treatment Response and Risk Stratification in Ewing Sarcoma. Clin Cancer Res [Internet]. 2021 Nov 15 [cited 2023 Feb 7];27(21):5922–30. Available from: https://pubmed.ncbi.nlm.nih.gov/34426444/

4. Meyer M, Seetharam M. First-Line Therapy for Metastatic Soft Tissue Sarcoma. Curr Treat Options Oncol [Internet]. 2019 Jan 1 [cited 2022 Oct 29];20(1). Available from: https://pubmed.ncbi.nlm.nih.gov/30675651/

5. Lee JH, Yoon YC, Seo SW, Choi Y La, Kim HS. Soft tissue sarcoma: DWI and DCE-MRI parameters correlate with Ki-67 labeling index. Eur Radiol [Internet]. 2020 Feb 1 [cited 2022 Jun 4];30(2):914–24. Available from: https://pubmed.ncbi.nlm.nih.gov/31630234/

6. Li X, Xie Y, Hu Y, Lu R, Li Q, Xiong B, et al. Soft tissue sarcoma: correlation of dynamic contrast-enhanced magnetic resonance imaging features with HIF-1α expression and patient outcomes. Quant Imaging Med Surg [Internet]. 2022 Oct 1 [cited 2023 Feb 14];12(10):4823–36. Available from: https://pubmed.ncbi.nlm.nih.gov/36185052/

---

## [Decision Letter · Decision Letter 1]

27 Apr 2023

Non-invasive monitoring of neoadjuvant radiation therapy response in soft tissue sarcomas by multiparametric MRI and quantification of circulating tumor DNA -- a study protocol

PONE-D-22-31540R1

Dear Dr. Runkel,

We’re pleased to inform you that your manuscript has been judged scientifically suitable for publication and will be formally accepted for publication once it meets all outstanding technical requirements.

Kind regards,

Alvaro Galli

Academic Editor

PLOS ONE

Additional Editor Comments (optional):

Reviewers' comments:

Reviewer's Responses to Questions

**Comments to the Author**

1. Does the manuscript provide a valid rationale for the proposed study, with clearly identified and justified research questions?

Reviewer #1: Yes

Reviewer #2: Yes

2. Is the protocol technically sound and planned in a manner that will lead to a meaningful outcome and allow testing the stated hypotheses?

Reviewer #1: Yes

Reviewer #2: Yes

3. Is the methodology feasible and described in sufficient detail to allow the work to be replicable?

Reviewer #1: Yes

Reviewer #2: Yes

4. Have the authors described where all data underlying the findings will be made available when the study is complete?

Reviewer #1: Yes

Reviewer #2: Yes

5. Is the manuscript presented in an intelligible fashion and written in standard English?

Reviewer #1: Yes

Reviewer #2: Yes

6. Review Comments to the Author

You may also provide optional suggestions and comments to authors that they might find helpful in planning their study.

Reviewer #1: The authors have answered all my questions in their response letter. I have no other concerns about this study.

Reviewer #2: Dear authors, thank you for the clarifications and good luck with your study. Please plan early enough the multicentric approach / validation.

7. PLOS authors have the option to publish the peer review history of their article (what does this mean?). If published, this will include your full peer review and any attached files.

Reviewer #1: No

Reviewer #2: No

---

## [Editor Report · Acceptance letter]

22 May 2023

PONE-D-22-31540R1 

Non-invasive monitoring of neoadjuvant radiation therapy response in soft tissue sarcomas by multiparametric MRI and quantification of circulating tumor DNA – a study protocol 

Dear Dr. Runkel:

I'm pleased to inform you that your manuscript has been deemed suitable for publication in PLOS ONE. Congratulations! Your manuscript is now with our production department. 

Kind regards, 

on behalf of

Dr. Alvaro Galli 

Academic Editor

PLOS ONE